# Immunogenicity and Safety of Half and Full Doses of Heterologous and Homologous COVID-19 Vaccine Boosters After Priming with ChAdOx1 in Adult Participants in Indonesia: A Single-Blinded Randomized Controlled Trial

**DOI:** 10.3390/vaccines13111149

**Published:** 2025-11-11

**Authors:** Nina Dwi Putri, Aqila Sakina Zhafira, Pratama Wicaksana, Hindra Irawan Satari, Eddy Fadlyana, Vivi Safitri, Nurlailah Nurlailah, Edwinaditya Sekar Putri, Nidya Putri, Devi Surya Iriyani, Yunita Sri Ulina, Frizka Aprilia, Evi Pratama, Indri Nethalia, Rita Yustisiana, Erlin Qur’atul Aini, Rini Fajarani, Adityo Susilo, Mulya Rahma Karyanti, Ari Prayitno, Hadyana Sukandar, Emma Watts, Nadia Mazarakis, Pretty Multihartina, Vivi Setiawaty, Krisna Nur Andriana Pangesti, Agnes Rengga Indrati, Julitasari Sundoro, Dwi Oktavia Handayani, Cissy B. Kartasasmita, Sri Rezeki Hadinegoro, Kim Mulholland

**Affiliations:** 1Faculty of Medicine, Universitas Indonesia, Dr. Cipto Mangunkusumo General Hospital, Jakarta 10430, Indonesiafrizkaaprilia@gmail.com (F.A.); indri.nethalia@petridish-id.org (I.N.); karyanti@ikafkui.net (M.R.K.);; 2Faculty of Medicine, Universitas Padjadjaran, Dr. Hasan Sadikin General Hospital, Bandung 40161, Indonesia; 3Tambora Primary Health Care Center, Jakarta 11320, Indonesia; 4Senen Primary Health Care Center, Jakarta 10410, Indonesia; 5Pasar Minggu Primary Health Care Center, Jakarta 12510, Indonesia; 6New Vaccines Research Group, Murdoch Children’s Research Institute (MCRI), Melbourne 3052, Australia; 7National Institute of Health Research & Development, Jakarta 10560, Indonesia; 8The Indonesian Technical Advisory Group on Immunization, Jakarta 10560, Indonesia; 9Jakarta Health Agency, Jakarta 10120, Indonesia

**Keywords:** COVID-19, vaccine, booster, half-dose

## Abstract

**Background**: Numerous studies have proved the efficacy of vaccination in reducing Severe Acute Respiratory Syndrome Coronavirus 2 (SARS-CoV-2) transmission and the coronavirus disease (COVID-19) burden. However, even though the COVID-19 vaccination coverage is high for primary doses, a booster dose is needed to sustain protection. Continuing our previous research, this study evaluates the immunogenicity and safety of full and half doses of two COVID-19 booster vaccines, ChAdOx1-S (AstraZeneca) and BNT162b2 (Pfizer-BioNTech), in individuals primed with ChAdOx1-S. **Methods**: This study was an observer-blind randomized controlled trial to evaluate the immunogenicity and safety of half and full doses of two COVID-19 booster vaccine types, BNT162b2 and ChAdOx1-S, among fully vaccinated, ChAdOx1-S-primed individuals in Jakarta, Indonesia. A total of 329 participants were randomized to receive either full or half doses of the booster vaccines, namely the ChAdOx1-S and BNT162b2 COVID-19 vaccines. Immunogenicity was assessed through SARS-CoV-2 antibody titers and neutralizing antibodies (NAbs) at 28 days post-booster, while safety was monitored via adverse event reporting. **Results**: The results showed that both vaccines demonstrated increased geometric mean titers (GMTs) post-booster. In the ChAdOx1-S booster group, at the baseline visit (day 0) and third visit (day 28), no statistically significant differences in GMT between the half- and full-dose groups were observed (*p* = 0.970 and 0.539, respectively). In the BNT162b2 group, no statistically significant difference was noted at the baseline visit, while the full dose was higher than the half dose at 28 days (Day 28, *p* = 0.011). Surrogate virus neutralization tests (sVNTs) and NAbs assays also revealed no significant differences between the half and full dose groups for both the Wuhan strain and the Delta variant. The BNT162b2 group compared to the ChAdOx1-S group revealed a statistically significant increase in IgG levels compared to ChAdOx1-S, with *p*-values of <0.001 and <0.001 for the half dose and full dose, respectively. This was also reflected in the NAbs test results, where BNT162b2 showed significantly higher levels against both the Wuhan strain and Delta variant. Adverse events were predominantly mild: 79.6% (*n* = 86/108) in the ChAdOx1-S full-dose group, 75.4% (*n* = 43/57) in the ChAdOx1-S half-dose group, 84.2% (*n* = 101/120) in the BNT162b2 full-dose group, and 92.6% (*n* = 88/95) in the BNT162b2 half-dose group, with pain at the injection site being the most common local reaction and myalgia and headache the most frequent systemic reactions. One serious adverse event was reported, assessed as unrelated to the vaccine. **Conclusions**: This study confirms that half doses of ChAdOx1-S and BNT162b2 are as immunogenic and safe as full doses, and a heterologous booster is more immunogenic than a homologous booster.

## 1. Introduction

Numerous studies have proven the efficacy of vaccination in reducing SARS-CoV-2 transmission and COVID-19 burden [1,2,3]. More than 20 COVID-19 vaccines have been approved for use, and more than 13.64 billion doses have been administered globally [4]. Now in the fifth year of the pandemic, Indonesia had a relatively high immunization coverage of 86.9% for the first dose and 74.6% for the second [5]. The most widely used primary series globally are BNT162b2 from Pfizer-BioNTech and ChAdOx1-S from AstraZeneca, followed by other vaccines, such as CoronaVac from Sinovac. However, at the time of the study, decision makers were considering options for booster doses to prolong protection at a time when vaccine supply was limited.

Most low-middle-income countries struggled to give adequate COVID-19 vaccines to their populations, making it appropriate to evaluate the alternative option of using lower doses in a greater number of people. In a study by Mahidol University in Thailand, a half dose (15 µg) of the BNT162b2 vaccine was compared with a full dose (30 µg) of BNT162b2 and with ChAdOx1-S as a booster in participants primed with ChAdOx1-S or Sinovac (CoronaVac). BNT162b2 immunogenicity measured with anti-SARS-CoV-2 RBD IgG, after two weeks, was comparable in those who had received CoronaVac prime or ChAdOx1-S half-dose prime. In the CoronaVac-primed group, neutralizing antibody (NAbs) responses in the half-dose BNT162b2 group were superior to those in the full-dose BNT162b2 group, a significant point for Indonesia since it supports the administration of a lower dose of BNT162b2, especially when the vaccine supply was limited [6]. In another booster study (CovBoost), which was carried out in the UK, half-dose (15 µg) or full-dose (30 µg) BNT162b2 was administered to participants primed with the BNT162b2 vaccine or ChAdOx1-S COVID-19 vaccines. On the 28-day post-booster evaluation using anti-spike IgG, both half and full dose groups showed acceptable immunogenicity and reactogenicity [7].

Hence, we proposed to examine booster vaccinations available to supplement the most commonly used primary series in Indonesia, including half doses when appropriate. The primary objective was to evaluate the immune responses to full and half booster doses for each COVID-19 vaccine tested.

This study is the same trial as BCOV21, a study on boosters in Indonesia. The study compared the immunogenicity of three vaccines (half and full doses of ChAdOx1-S and BNT162b2 and a full dose of CoronaVac) on participants who had received the CoronaVac primary series. All groups achieved seropositivity at 28 days post-inoculation [8]. This paper focuses on ChAdOx1-S-primed individuals.

We describe here an observer-blinded randomized controlled trial to assess the immunogenicity and safety of two potential booster vaccines (ChAdOx1-S and BNT162b2) at full or half doses in Indonesian participants fully vaccinated with the second most commonly used primary series in Indonesia, ChAdOx1-S. This study was conducted to provide information directly applicable to the authorities and community. These data were used by the Ministry of Health in devising the 2022 policy regarding booster doses and schedules for the general public.

## 2. Materials and Methods

### 2.1. Study Design, Participants, and Ethics

This study was conducted under the same protocol as the BCOV21 study [8]. We aimed to conduct an observer-blind randomized controlled trial to evaluate the immunogenicity and safety of a full dose or half dose of two COVID-19 booster vaccines (BNT162b2 and ChAdOx1-S) in healthy adults above 18 years old who had been fully vaccinated with ChAdOx1-S within 6–9 months before enrollment in Indonesia (Appendix Table A1). This study was conducted in two Primary Health Care Centers (PHCs) in Jakarta, which are PHC Senen and PHC Tambora. In this component of the study, we recruited 329 participants who had received full primary doses of ChAdOx1-S and signed the informed consent form. The participants were enrolled from January to March 2022.

The BCOV21 study previously reported the results of booster responses among participants who had received CoronaVac as their primary series. The current paper represents a separate subset of participants from the same protocol, focusing specifically on those primed with ChAdOx1-S and evaluating the immunogenicity and safety of BNT162b2 and ChAdOx1-S boosters. Therefore, all immunogenicity, neutralizing antibody, and reactogenicity data presented here are new and distinct from those included in the prior BCOV21 publication.

This study was conducted according to the Declaration of Helsinki and the International Conference on Harmonization—Good Clinical Practice guidelines. It was approved by the Health Research Ethics Committee and the National Drug and Food Authority, Indonesia.

### 2.2. Trial Protocol

A total of three visits were planned for each participant. At the baseline visit (Day 0), following the delineation of the study course, informed consent was sought from all participants to be included in the study. Study doctors then reviewed the inclusion and exclusion criteria based on the protocol. Before vaccine booster administration, blood sampling was performed on all eligible participants for the pre-vaccination immunogenicity outcome. After receiving the vaccine, the participants were given a diary card to record any adverse events and concomitant medication usage during the study period. The second visit (Day 7) was conducted 7 days after booster vaccination to follow up on adverse events, concomitant medication, and if they had any COVID-19 symptoms. At the third visit (Day 28), 1 month after booster administration, blood samples were obtained for immunogenicity testing.

All participants with COVID-19 symptoms were reported and treated based on the Indonesian COVID-19 Task Force regulations [9].

### 2.3. Randomization and Blinding

The unblinded study team randomly assigned eligible participants to the randomization group. The study used stratified randomization with separate randomization lists in each primary health center according to age group (18–59 and >60 years old) with a ratio of 4:1. Every participant was randomized to receive either a half or full dose of ChAdOx1-S, BNT162b2, or a full dose of CoronaVac intramuscularly (ChAdOx1-S 0.5 mL or 0.25 mL, BNT162b2 0.3 mL or 0.15 mL, or CoronaVac 0.5 mL). However, in this paper, we only present the results of the BNT162b2 and ChAdOx1-S booster vaccines, as the first part of the BCOV21 investigation in CoronaVac prime doses participants found an inadequate immune response in the CoronaVac-booster group [8]. The exclusion of the CoronaVac booster arm was a post hoc decision based on these findings.

### 2.4. Outcomes

Immunogenicity was the primary outcome of this study, comprising the SARS-CoV-2 binding antibody titers and NAbs at 28 days after booster dose vaccination. All samples were tested for the cumulative titer of IgG anti-S-RBD of SARS-CoV-2 antibody using the Abbott SARS-CoV-2 IgG II Quant assay (Abbott Laboratories, Abbott Park, IL, USA), a chemiluminescent microparticle immunoassay from Abbott Laboratories [10]. Results were expressed in Arbitrary Units per milliliter (AU/mL) and converted to Binding Antibody Units per milliliter (BAU/mL) using the manufacturer’s conversion factor (AU/mL × 0.142 = BAU/mL) to allow inter-study comparability with other standardized serological assays [11].

Seropositivity was defined as an IgG antibody concentration of at least 7.1 BAU/mL. Seroconversion in participants with pre-booster IgG antibody levels of 7.1 BAU/mL was defined as a ≥4-fold increase in antibody concentration 28 days post-booster compared to baseline. For participants with pre-booster IgG levels below 7.1 BAU/mL, seroconversion was defined as reaching antibody titers of ≥7.1 BAU/mL [12]. A subset of samples was identified by randomization to be analyzed for neutralizing antibody levels: 60 samples per randomization group were tested with the cPass™ surrogate virus neutralization test (sVNT) Kit against the Wuhan and Delta variants, and 10 samples per randomization group were analyzed with the Conventional Antibody Neutralization Test. Seroconversion in the surrogate virus neutralization test (sVNT) was defined by >30% inhibition for the detection of antibodies against SARS-CoV-2 [13]. Samples collected were processed onsite and sent directly to the National Institute of Health Research and Development, Indonesia. The secondary outcome was the reactogenicity after booster administration. The incidence of adverse events, followed by rate intensity, was observed 24 h, 7 days, and 28 days after booster dose administration.

### 2.5. Statistical Analysis

The sample size for the immunogenicity analysis was determined based on the assumption that at least 95% of participants would achieve seropositivity 28 days after the booster dose. Using a hypothesized seropositive proportion of 0.95, an anticipated proportion of 0.85, a two-sided significance level of 0.05, and a statistical power of 80%, the calculation yielded a requirement of 53 participants per intervention group. After accounting for a 10% dropout rate, approximately 60 participants per group were needed for immunogenicity analysis. To ensure sufficient power for both immunogenicity and reactogenicity endpoints, the final sample size was set at 100 participants per intervention group, providing 80% power at a 5% significance level to detect an absolute difference of 10–15% in seropositivity between groups.

The research data were processed and analyzed descriptively and analytically. The descriptive analysis presented percentages with 95% confidence intervals for categorical data and logarithmic means with 95% confidence intervals for immunogenicity data. Regarding statistical tests, the chi-square test was used to compare the proportions of the four groups’ administration of booster doses following a primary dose of ChAdOx1-S, the Mann–Whitney test to compare continuous variables between groups, and the geometric mean titer (GMT) of the four groups with a primary series of the ChAdOx1-S vaccine. Statistical analysis was performed using IBM SPSS Statistics software version 18. A *p*-value of < 0.05 was considered statistically significant.

## 3. Results

Between the 28 January 2022 and the 25 March 2022, 340 participants signed the Informed Consent Form (ICF) and were screened (Figure 1). Eleven participants did not meet eligibility criteria. As a result, a total of 329 participants were enrolled. All participants had ChAdOx1-S as the primary series and were randomized to receive a unique study identification number and randomization code and receive one of the booster vaccine products and doses provided. Baseline characteristics were similar between all groups (Table 1). The CoronaVac full-dose booster arm was excluded from this analysis due to previously reported insufficient immune responses and limited evaluable data.

At the baseline visit in the ChAdOx1-S and BNT162b2 groups, there was no significant difference between those who were subsequently given full or half doses at day 0. After the booster at day 28, there was no difference between the responses to ChAdOx1-S full and half doses (Figure 2). The GMT was 471.18 BAU/mL (95% CI 396.35–560.13) for the half-dose group and 531.56 BAU/mL (95% CI 455.83–619.87) for the full-dose group (*p* = 0.539), with a geometric mean ratio (GMR) of 1.02 (95% CI 0.78–1.34). The mean difference between groups was 53.78 BAU/mL (95% CI −93.9 to 201.47), and the effect size (Cohen’s d = 0.263) indicated a small and clinically negligible difference.

There was, however, a significant difference in GMT between the BNT162b2 full and half doses between the groups at the day 28 time point (*p* = 0.011). The GMTs were 1574.48 BAU/mL (95% CI 1362.66–1819.23) for the half-dose group and 2060.147 BAU/mL (95% CI 1734.98–2331.56) for the full-dose group, corresponding to a GMR of 1.19 (95% CI 0.75–1.48). The mean difference was 460.39 BAU/mL (95% CI 88.54–832.24) with a small-to-moderate effect size (Cohen’s d = 0.377).

The increase in IgG anti-S-RBD titer after day 28 in the ChAdOx1-S group was 111.61 BAU/mL (−1715.33–3204.32) and 134.27 BAU/mL (−598.40–2962.12) for half and full dose, respectively. The increase in BNT162b2 booster vaccines was 1137.76 BAU/mL (56.16–5165.57) and 1677.11 BAU/mL (26.66–5620.97) for half and full doses, respectively (Appendix Table A2). This shows that BNT162b2 has a statistically higher value, with a *p*-value of <0.001 for both half and full doses.

For the sVNT against the Wuhan strain antigen, at the day 0 time point, no statistically significant difference between half-dose and full-dose groups was noted, with 42 participants (93.3%) and 47 (95.9%) testing positive for NAbs, respectively (*p* = 0.537) (Appendix Table A3). There was also no statistically significant difference between the groups at the day 28 time point, with 45 participants (100%) positive in the half-dose group and 49 participants (100%) positive in the full-dose group. For the sVNT against the Delta variant antigen, at the day 0 time point, prior to ChAdOx1-S administration, no statistically significant difference was seen between the half-dose and full-dose groups, with 42 participants (93.3%) and 49 (100%) testing positive for NAbs, respectively (*p* = 0.188). There was also no statistically significant difference between the groups at the day 28 time point, with 45 participants (100%) positive in the half-dose group and 49 participants (100%) positive in the full-dose group (*p* = 0.306).

For the Wuhan strain antigen, at the Day 0 time point, no significant difference was seen between the half-dose and full-dose groups, with 49 participants (96.1%) and 47 (100%) testing positive, respectively (*p* = 0.292) (Appendix Table A4). There was also no significant difference between the groups at the day 28 time point, with 51 participants (100%) positive in the half-dose group and 47 participants (100%) positive in the full-dose group (*p* = 0.351). For the sVNT Delta variant antigen, at the day 0 time point, prior to BNT162b2 administration, no significant difference was seen between the half-dose and full-dose groups, with 46 (92%) and 46 participants (97.9%) testing positive in the two groups, respectively (*p* = 0.301). There was also no statistically significant difference between the groups at the day 28 time point, with 48 participants (96%) and 47 participants (100%) positive in the half-dose and full-dose groups, respectively (*p* = 0.115).

We divided the analysis into day 0 and day 28 time points for the Wuhan strain and Delta variant antigen, both with ChAdOx1-S vaccine priming. For the Wuhan strain antigen, at the day 0 time point, there was no statistically significant difference between the half and the full ChAdOx1-S booster dose groups, with GMTs of 94.06 dilution titer (28.48–310.67) and 203.19 dilution titer (31.06–1329.23), respectively (*p* = 0.257). At the day 28 time point, there was also no significant difference between the half-dose and full-dose groups, with GMTs of 128.0 dilution titer (63.26–259.00) and 181.02 dilution titer (27.39–876.19), with a *p*-value of 0.391. The GMI increase for the half-dose group was 1.36 dilution titer (0.74–2.48) and 0.89 dilution titer (0.38–2.08) for the full-dose group. For the delta variant antigen, at the day 0 time point, the GMT for the half-dose group was 59.26 dilution titer (64, 95% CI 27.32–128.51), and that for the full-dose group was 143.68 dilution titer (95% CI 49.24–419.17). There was no statistically significant difference between the two groups (*p* = 0.133). At the day 28 time point, the GMT for the half-dose group was 80.63 dilution titer (64, 95% CI 47.33–137.38), and that for the full-dose group was 143.68 dilution titer (95% CI 49.24–419.17), with no statistically significant difference between the groups (*p* = 0.227). The GMI was 1.36 dilution titer (0.85–2.18) for the half-dose group and 1.00 dilution titer (0.63–1.58) for the full-dose group.

Regarding the NAbs results at day 0 and day 28 for half and full doses of the BNT162b2 COVID-19 vaccine, the analysis considered the Wuhan strain antigen and the delta variant antigen, both with ChAdOx1-S vaccine priming (Figure 3). For the Wuhan strain antigen at the day 0 time point, there were no statistically significant differences between the half-dose and full-dose groups, with GMTs of 147.03 dilution titer (88.10–245.36) and 109.73 dilution titer (40.82–295.98), respectively (*p* = 0.736). At the day 28 time point, there were also no statistically significant differences between the half-dose and full-dose groups, with GMTs of 477.71 dilution titer (242.16–942.54) and 552.99 dilution titer (297.03–1029.67), respectively (*p* = 0.769). The GMIs are highly significant, with the increasing number for the half-dose group being 3.25 dilution titer (95% CI 1.92–5.49) and the full-dose group being 5.04 dilution titer (2.00–12.68). For the delta variant antigen, at the day 0 time point, there was no statistically significant difference between the two groups (*p* = 0.676). At the day 28 time point, the GMT for the half-dose group was 207.94 dilution titer (148.79–290.59), and that for the full-dose group was 276.50 dilution titer (182.30–419.35), with no statistically significant difference between groups (*p* = 0.233). The GMI for the half-dose group was 2.64 dilution titer (1.55–4.50), and that for the full-dose group was 2.72 dilution titer (1.00–7.40). When comparing the increase in NAbs, we found that there was no increase in the ChAdOx1-S booster group. In contrast, the BNT162b2 booster produced statistically higher neutralizing antibody results compared to the ChAdOx1-S booster dose in both the Wuhan strain (half dose *p*-value: 0.006; full dose *p*-value: 0.002) and the Delta variant (half dose *p*-value: 0.001), except in the full dose group for the Delta variant, where the *p*-value was 0.224.

This study evaluated the adverse events (AEs) after administering the full and half doses of the ChAdOx1-S COVID-19 vaccine and the BNT162b2 COVID-19 vaccine. Figure 4 and Figure 5 show the severity level, duration, and type of local and systemic adverse events after booster vaccination for each vaccine. 

In this study, a total of 156 local AEs and 224 systemic AEs were reported. Most of the local adverse events involved local pain, which the AE in both full-dose groups showed greater effect (23.07%, *n* = 36 and 23.71%, *n* = 37 in ChAdOx1-S and BNT162b2, respectively). Other common local adverse events were swelling, induration, and redness. On the other hand, the most prevalent systemic adverse events were myalgia (5.8%, *n*= 13/224) in the ChAdOx1-S full-dose group, fatigue and headache (both 6.69%, *n* = 15/224) in the BNT162b2 full-dose group, and headache (8.26%, *n* = 19/224) in the BNT162b2 half-dose group. Across all groups, local adverse events occurred at an overall incidence of 1.69 per 100 person-days, while systemic adverse events occurred at 2.43 per 100 person-days. Incidence rates were comparable between full- and half-dose groups for both ChAdOx1-S and BNT162b2.

Only one grade 4 adverse event (1.8%, *n* = 4/224) occurred in the ChAdOx1-S half-dose group, with high fever (>40 °C) occurring 1 day after vaccination. Most of the participants experienced events of grade 1 severity, numbering 79.6% (*n* = 86/108) in the ChAdOx1-S full-dose group, 75.4% (*n* = 43/57) in the ChAdOx1-S half-dose group, 84.2% (*n* = 101/120) in the BNT162b2 full-dose group, and 92.6% (*n* = 88/95) in the BNT162b2 half-dose group. Most of the participants with adverse events experienced the adverse event for 7 days: 70.4% (*n* = 76/108) in the ChAdOx1-S full-dose group, 70.2% (*n* = 40/57) in the ChAdOx1-S half-dose group, 82.5% (*n* = 99/120) in the BNT162b2 full-dose group, and 89.5% (*n* = 85/95) in the BNT162b2 half-dose group.

One participant had a serious adverse event. This participant, who was enrolled in the ChAdOx1-S half-dose group, was hospitalized for pneumonia and uncontrolled diabetes mellitus. The investigator considered that the event was not related to the booster vaccine because of the lengthy time (13 days) between administering the booster vaccine and the serious adverse event. It is also based on the Data and Safety Monitoring Board (DSMB) review.

## 4. Discussion

This study evaluates the effects of half-dose and full-dose heterologous and homologous boosters given to participants who received the primary ChAdOx1-S dose. The immunogenicity results, measured by IgG levels, indicate that the half dose produces the same immunogenic response as the full dose. Neutralizing antibody results further support these findings. The heterologous approach demonstrates higher antibody levels when comparing heterologous and homologous boosters. Additionally, the evaluation of reactogenicity revealed that a higher number of participants in the full-dose group reported adverse events compared to those who received the half-dose. Furthermore, this study showed that all participants who received a ChAdOx1-S primary series still had adequate binding antibody levels 6–9 months after their second dose, with only 4 participants (1.2%) showing low values.

The IgG results from our study were consistent with a recent study from the UK stating that ChAdOx1-S vaccination led to increased levels of total IgG antibodies following a third dose [14]. In Cyprus, seropositivity rates and antibody titers were higher for BNT162b2 and ChAdOx1-S vaccines than for CoronaVac in all age groups at all time points evaluated. Comparing the results of full and half doses, it is evident that although the GMT is lower for half doses, the difference is not statistically significant. Importantly, interpretation of the confidence intervals around these estimates showed substantial overlap between the half- and full-dose groups, suggesting comparable immunogenicity while acknowledging that the study was not designed to formally test non-inferiority. This use of confidence intervals provides a more informative understanding of the precision of the estimates than reliance on *p*-values alone.

This finding aligns with other studies, which have also shown no significant difference between half-dose and full-dose BNT162b2 booster vaccines [15]. Additionally, this study compared heterologous boosters to homologous boosters, revealing that the results from heterologous boosters were significantly higher than those from homologous boosters. While there is no specific threshold for protection, these findings are consistent with other research in the field [15].

We also analyzed the sVNT at the day 0 and day 28 time points using the Delta variant and Wuhan strain antigens. The results revealed no statistically significant difference between the half-dose and full-dose groups at these time points, both for the Delta variant and the Wuhan strain antigens of the BNT162b2 and ChAdOx1-S vaccines. This finding was similar to study results from Thailand and the UK, which revealed strong antibodies against Omicron and four other variants, including Delta, after a third dose of an mRNA vaccine [6,7]. Meanwhile, a ChAdOx1-S heterologous boost was more immunogenic in CoronaVac-prime recipients than in ChAdOx1-S-prime recipients [6]. Along with the sVNT, we analyzed NAbs at the day 0 and day 28 time points for both the Wuhan strain and the Delta variant antigen. The results also revealed no statistically significant difference between the half-dose and full-dose groups at these time points for either antigen with the ChAdOx1-S and BNT162b2 vaccines.

Following the booster injections, we examined AEs and SAEs throughout the observation period. Most participants experienced events of grade 1 (mild) severity during the 7 days following immunization with both vaccines. There were differences in the ChAdOx1-S group, in which the participants who received the full dose of the vaccine had more local adverse events than those given the half dose. In the BNT162b2 groups, individuals who were given the full dose of the vaccine also had more local adverse events. Additionally, participants who received the half dose of the ChAdOx1-S booster had fewer local adverse reactions than those who received the half dose of BNT162b2. This is consistent with other studies showing mRNA vaccinations to be the most reactogenic, followed by viral vector vaccines and protein subunit vaccines, while inactivated vaccines were the least reactogenic [16]. There were no significant differences in the distribution of systemic adverse events for both vaccines. These findings were in line with a study in Thailand, which showed that heterologous booster immunization was usually well-tolerated, and the rates of adverse events were consistent with those reported in primary and booster COVID-19 vaccine studies [6].

This study has some limitations. The sample size in this study was insufficient to assess rare vaccine side effects, even though no tolerability issues were reported in those who received ChAdOx1-S or BNT162b2, half dose or full dose, and the small sample size in the CoronaVac group limited our ability to evaluate the immunogenicity and safety of the CoronoVac as the booster dose. Second, the follow-up duration was short, making it impossible to determine the long-term protection provided by the vaccines and to study the breakthrough infection. However, the study is ongoing, and we examined the SARS-CoV-2 antibody titers at various time points. Third, only a subset of 10 samples per group was analyzed using the conventional neutralizing antibody (NAb) assay due to limited laboratory resources. Nevertheless, the NAb results were consistent with those obtained from the surrogate virus neutralization test (sVNT), supporting the validity of the findings despite the smaller subsample. Fourth, our analysis focused solely on IgG data related to the Wuhan strain, without including the Delta variant or other emerging variants. Finally, since the study mainly included healthy adult participants in Indonesia, the findings may be inapplicable to other populations, such as immunocompromised participants.

This study found no significant differences in either immunogenicity or reactogenicity between half- and full-dose booster vaccinations. It proved that administering half and full doses of heterologous and homologous COVID-19 vaccine boosters is safe and may provide adequate protection against SARS-CoV-2 infection, with heterologous vaccines showing better immunogenicity results. Our findings suggest that administering lower doses of BNT162b2 and ChAdOx1-S booster vaccines could assist countries with limited vaccine supplies to achieve the same level of immunogenicity. Additionally, reducing the dose may decrease adverse reactions, thereby improving vaccine uptake.

## 5. Conclusions

This study confirms that-half dose boosters of ChAdOx1-S and BNT162b2 are as immunogenic and safe as full-dose regimens. Heterologous booster vaccination induced stronger immune responses than homologous boosting, with all regimens demonstrating good tolerability and predominantly mild adverse events. These findings support the use of half-dose and heterologous booster strategies as efficient and safe alternatives, particularly in settings with limited vaccine availability. Continued monitoring is needed to evaluate the durability of immune protection over time.

## Figures and Tables

**Figure 1 vaccines-13-01149-f001:**
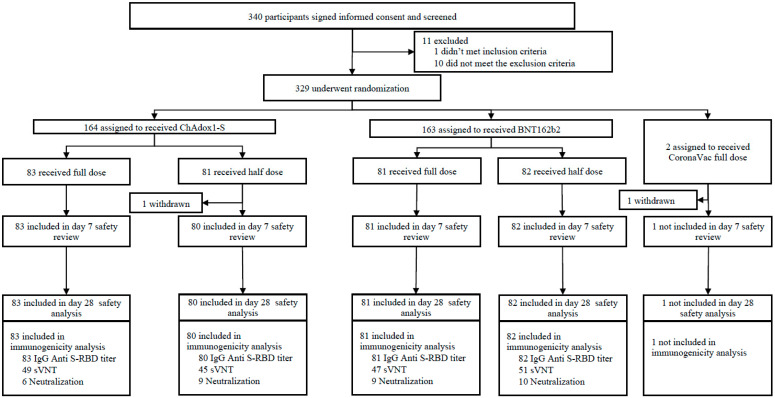
Trial profile.

**Figure 2 vaccines-13-01149-f002:**
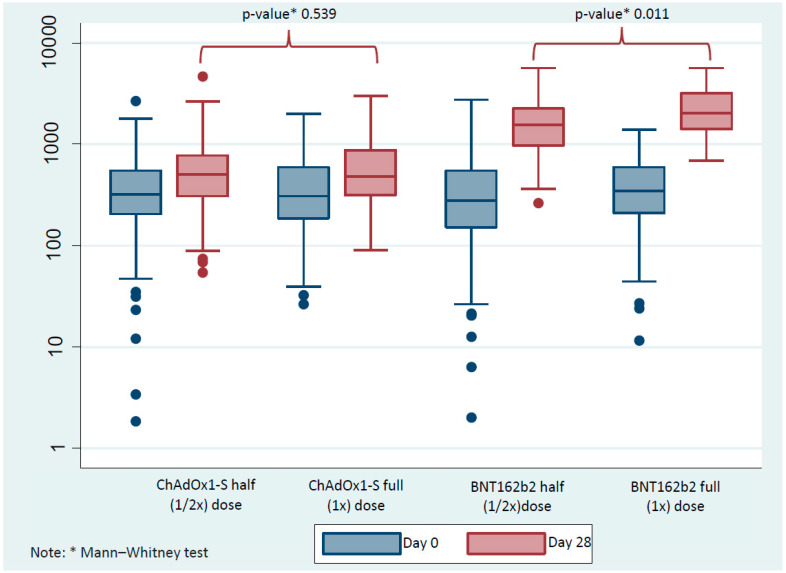
The difference in immunoglobulin G (IgG) anti-spike protein receptor-binding domain (anti-S-RBD) (BAU/mL) as evaluated at day 0 and day 28 between half (1/2×) and full (1×) doses of the ChAdOx1-S and BNT162b2. A Mann–Whitney test was used to compare between half and full doses at each time point. Solid blue and red circles indicate outliers.

**Figure 3 vaccines-13-01149-f003:**
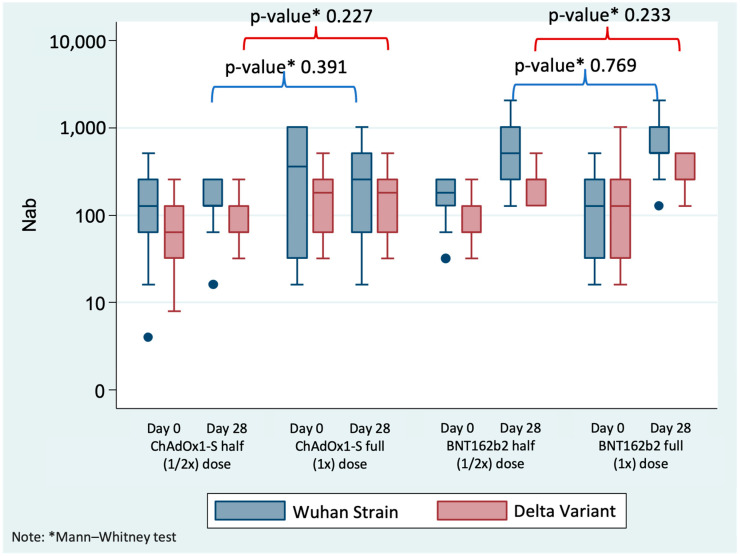
Shows the Nabs (dilution titer) results at day 0 and day 28 for half (1/2×) and full (1×) doses of the ChAdOx1-S COVID-19 vaccine (ChAdOx1-S). A Mann–Whitney test was used to compare half and full dose groups at each time point. Solid blue circles indicate outliers.

**Figure 4 vaccines-13-01149-f004:**
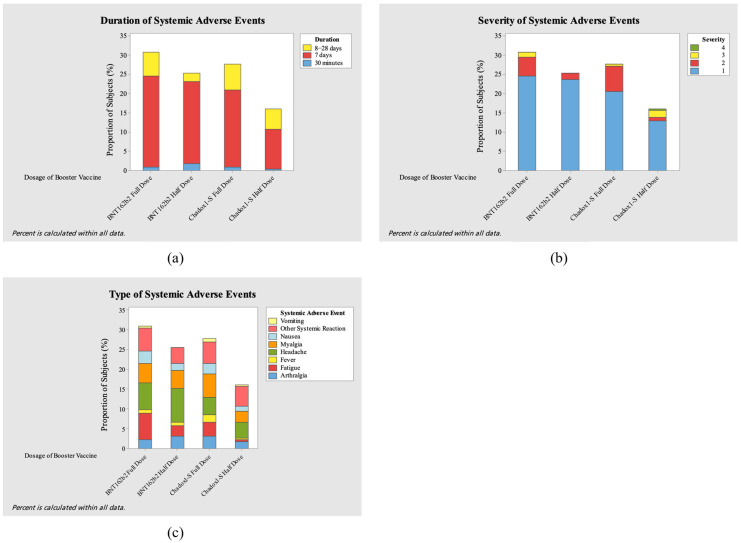
Severity (**a**), duration (**b**), and type (**c**) of local adverse events (AEs) after booster vaccination.

**Figure 5 vaccines-13-01149-f005:**
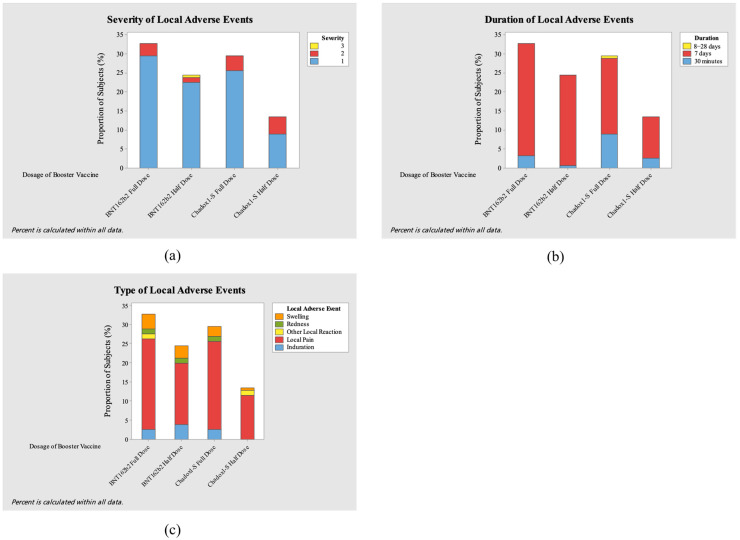
Severity (**a**), duration (**b**), and type (**c**) of systemic adverse events (AEs) after booster vaccination.

**Table 1 vaccines-13-01149-t001:** Demographic data.

Parameter	ChAdOx1-S Full Dose	ChAdOx1-S Half Dose	BNT162b2 Full Dose	BNT162b2 Half Dose	CoronoVac Full Dose
	(*n* = 83)	(*n* = 81)	(*n* = 81)	(*n* = 82)	(*n* = 2)
Means age (years) (SD)	42.60 (11.67)	44.7 (10.97)	41.50 (10.54)	41.95 (12.13)	40.10 (19.65)
Means height (cm) (SD)	158.5 (9.17)	157.80 (8.68)	156.00 (8.01)	155.50 (8.80)	162.25 (13.78)
Means weight (kg) (SD)	63.90 (13.57)	62.80 (15.11)	64.00 (13.50)	63.45 (13.52)	65.30 (13.57)
BMI (kg/m2)	25.50 (5.18)	25.20 (5.24)	25.90 (4.85)	26.65 (5.17)	24.65 (0.91)
Sex *n* (%)					
Male	38 (25.0%)	43 (28.3%)	34 (22.4%)	36 (23.7%)	1 (0.7%)
Female	45 (25.4%)	38 (21.5%)	47 (26.6%	46 (26.0%)	1 (0.6%)

SD: Standard Deviation.

## Data Availability

All de-identified participant data are available upon request. Interested parties may contact the corresponding author at ninadwip@gmail.com with a formal proposal and a signed data access agreement. Upon approval by the study team, sponsor, and collaborators, the data will be shared via a secure online platform.

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
