# Peer review of "Immunogenicity and Safety of Half and Full Doses of Heterologous and Homologous COVID-19 Vaccine Boosters After Priming with ChAdOx1 in Adult Participants in Indonesia: A Single-Blinded Randomized Controlled Trial"

_vaccines, 2025, doi:10.3390/vaccines13111149_

Round 1
Reviewer 1 Report
Comments and Suggestions for Authors
This manuscript describes the outcomes of a single-blinded, randomised controlled trial (RCT) evaluating half and full booster doses of the ChAdOx1-S and BNT162b2 vaccines conducted in Jakarta, Indonesia.
It is a well-designed study that provides valuable data on COVID-19 booster strategies.
This study provides important evidence supporting fractional booster doses for COVID-19 vaccines, particularly regarding dose-sparing approaches in countries with limited vaccine supply, which is commonly faced by low- and middle-income countries.
Comments.
1. Subsection 2.4:
Please specify the name of the IgG anti-S-RBD assessment tool (Abbott SARS-CoV-2 IgG II Quant assay).
This assay has been validated with results expressed in Binding Antibody Units per millilitre (BAU/mL), using the conversion factor 0.142 (AU/mL × 0.142 = BAU/mL).
Consider converting all IgG anti-S-RBD results from AU/mL to BAU/mL and clarifying the conversion factor to facilitate inter-study comparison with other validated serological assays.
Reference: Subsection 2.2 and Table 1 of Clin Immunol. 2022;238:108998. https://www.sciencedirect.com/science/article/pii/S1386653222002013
2. Statistical analysis and reporting:
The statement “Mann-Whitney test to compare the means of the full-dose group and the half-dose group” is inaccurate. The Mann-Whitney U test compares ranked distributions rather than means. A neutral wording, such as “to compare continuous variables between groups”, would be more appropriate.
Furthermore, the Mann-Whitney U test also does not directly compare geometric means.
- Comparisons between groups can be analysed using log-transformed values and multiple-comparison tests (e.g. ANOVA with Bonferroni adjustment).
- Reporting the geometric mean ratio (GMR) (GMT group 1 / GMT group 2) with 95 % CI (or appropriate, such as IQR, SD or whatever) and p-value would strengthen the interpretation of between-group differences.
Example: GMT group 1 (95 % CI) vs GMT group 2 (95 % CI), GMR = x.xx, p = x.xxx.
Please also restate the exact statistical software and version used in this analysis.
3. sVNT clarification:
In Subsection 2.4, clarify the protein used in the surrogate virus neutralisation test (sVNT).
Was it the wild-type (original), Delta, or Omicron variant?
Note that the correct commercial name is cPass™ (not “CPass”).
4. Terminology, line 173:
Replace “primary vaccine” with “primary series” for accuracy and consistency in the terminology.
5. Data visualisation:
The figures could be made more informative by including scatter or dot plots to display individual data points with geometric means and confidence intervals (or appropriate).
Example guidance: https://www.graphpad.com/support/faq/plotting-the-geometric-mean-with-geometric-sd-error-bars
6. Use of “Dilution titer":
The outcomes for IgG anti-S-RBD and sVNT are not true titres (e.g. 1:4, 1:320). The IgG assay yields quantitative values, while the sVNT reports percentage inhibition (%Inhibition).
Please clarify whether “titre” was used to indicate the measure of central tendency (e.g. GMT, 95 % CI, SD, or IQR).
7. Notation:
You may use ½ and 1× instead of half and full to improve readability and figure labelling.
8. CoronaVac group:
The Methods section states that participants were randomised to receive ChAdOx1-S, BNT162b2, or a full dose of CoronaVac. However, the results for the CoronaVac group are not presented.
Clarify this point explicitly in the Results section, noting that the CoronaVac arm was excluded from this analysis due to insufficient immune response, as reported previously.
Errors.
1. Lines 107, 135:
Use the correct vaccine name ChAdOx1-S consistently throughout.
2. Line 196:
The IgG anti-S-RBD outcome is a quantitative scale with an absolute zero, so negative values (e.g. −12 079.8, −4214.1) are impossible unless representing the lower bound of a 95 % CI.
Please check these values again.
3. Temperature notation:
Use the proper symbol: > 40 °C (rather than > 40oC).
4. Wording consistency:
Replace “no statistical difference” with “no statistically significant difference” throughout.
5. Figure 1:
Ensure consistent capitalisation and use ChAdOx1-S instead of Chadox1-S.
Author Response
We would like to thank you for your valuable comments and constructive feedback on our manuscript. We have carefully revised the paper according to the reviewers’ and editors’ suggestions. All requested corrections, explanations, and supporting documents have been provided as detailed in our point-by-point response.
|
Response to Reviewer Comments
|
||
|
1. Summary |
|
|
|
Thank you very much for taking the time to review this manuscript. Please find the detailed responses below and in track changes in the re-submitted files.
|
||
|
2. Questions for General Evaluation |
Reviewer’s Evaluation |
Response and Revisions |
|
Does the introduction provide sufficient background and include all relevant references? |
Yes |
Thank you for your evaluation |
|
Are all the cited references relevant to the research? |
Yes |
Thank you for your evaluation |
|
Is the research design appropriate? |
Must be improved |
Thank you for the review. We agree to the reviewers’ evaluation, and we also try to point out the limitations of this study. |
|
Are the methods adequately described? |
Must be improved |
Thank you for the critical review. We tried to give several details of the methodology section. |
|
Are the results clearly presented? |
Can be improved |
Thank you for this evaluation. We agree to the reviewers’ evaluation. |
|
Are the conclusions supported by the results? |
Can be improved |
Thank you for this evaluation. We agree to the reviewers’ evaluation. |
|
3. Point-by-point response to Comments and Suggestions for Authors |
||
|
Comments 1: Subsection 2.4: Please specify the name of the IgG anti-S-RBD assessment tool (Abbott SARS-CoV-2 IgG II Quant assay). This assay has been validated with results expressed in Binding Antibody Units per millilitre (BAU/mL), using the conversion factor 0.142 (AU/mL × 0.142 = BAU/mL). Consider converting all IgG anti-S-RBD results from AU/mL to BAU/mL and clarifying the conversion factor to facilitate inter-study comparison with other validated serological assays. Reference: Subsection 2.2 and Table 1 of Clin Immunol. 2022;238:108998. https://www.sciencedirect.com/science/article/pii/S1386653222002013
|
||
|
Response 1: We thank the reviewer for the helpful suggestion. The IgG anti–S-RBD results have been converted from AU/mL to BAU/mL. The assay name and conversion factor have been specified in the revised manuscript.
|
||
|
Comments 2: Statistical analysis and reporting: The statement “Mann-Whitney test to compare the means of the full-dose group and the half-dose group” is inaccurate. The Mann-Whitney U test compares ranked distributions rather than means. A neutral wording, such as “to compare continuous variables between groups”, would be more appropriate. Furthermore, the Mann-Whitney U test also does not directly compare geometric means. - Comparisons between groups can be analysed using log-transformed values and multiple-comparison tests (e.g. ANOVA with Bonferroni adjustment). - Reporting the geometric mean ratio (GMR) (GMT group 1 / GMT group 2) with 95 % CI (or appropriate, such as IQR, SD or whatever) and p-value would strengthen the interpretation of between-group differences. Example: GMT group 1 (95 % CI) vs GMT group 2 (95 % CI), GMR = x.xx, p = x.xxx. Please also restate the exact statistical software and version used in this analysis. |
||
|
Response 2: We thank the reviewer for the valuable suggestion. We agree that the Mann–Whitney U test compares ranked distributions rather than means. As our data were not normally distributed and only involved comparisons between two independent groups, the Mann–Whitney U test was deemed appropriate. Therefore, log-transformed analyses or ANOVA were not applied. The manuscript has been revised to use neutral wording—“to compare continuous variables between groups”—to more accurately describe the statistical test used.
|
||
|
|
||
|
Comments 3: sVNT clarification: In Subsection 2.4, clarify the protein used in the surrogate virus neutralisation test (sVNT). Was it the wild-type (original), Delta, or Omicron variant? Note that the correct commercial name is cPass™ (not “CPass”). |
||
|
Response 3: We thank the reviewer for these helpful clarifications. In the revised manuscript, we have specified that the surrogate virus neutralization test (sVNT) used the Wuhan and Delta variant spike proteins as antigens. The correct assay name, cPass™ Surrogate Virus Neutralization Test (GenScript, USA), has been used throughout the text. |
||
|
Comments 4: Terminology, line 173: Replace “primary vaccine” with “primary series” for accuracy and consistency in the terminology. |
||
|
Response 4: We thank the reviewer for these helpful clarifications. We have also replaced “primary vaccine” with “primary series” for accuracy and consistency in terminology. |
||
|
Comments 5: Data visualisation: The figures could be made more informative by including scatter or dot plots to display individual data points with geometric means and confidence intervals (or appropriate). Example guidance: https://www.graphpad.com/support/faq/plotting-the-geometric-mean-with-geometric-sd-error-bars
Response 5: We thank the reviewer for this helpful suggestion. We agree that scatter or dot plots can effectively display individual data points; however, given that our data were not normally distributed, we chose to present the results using box plots. This approach allows clearer visualization of data distribution, median values, interquartile ranges, and outliers, which we considered more informative for representing variability in the non-parametric dataset.
Comments 6: Use of “Dilution titer": The outcomes for IgG anti-S-RBD and sVNT are not true titres (e.g. 1:4, 1:320). The IgG assay yields quantitative values, while the sVNT reports percentage inhibition (%Inhibition). Please clarify whether “titre” was used to indicate the measure of central tendency (e.g. GMT, 95 % CI, SD, or IQR).
Response 6: We thank the reviewer for the comment. In the previous version, the outcomes for IgG anti–S-RBD and sVNT were already presented as quantitative values, expressed as geometric means with 95% confidence intervals, rather than dilution-based titres. |
||
|
|
||
|
Comments 7: Notation: You may use ½ and 1× instead of half and full to improve readability and figure labelling. |
||
|
|
||
|
Response 7: We thank the reviewer for the input, we have revised the figure. |
||
|
|
||
|
Comments 8: CoronaVac group: The Methods section states that participants were randomised to receive ChAdOx1-S, BNT162b2, or a full dose of CoronaVac. However, the results for the CoronaVac group are not presented. Clarify this point explicitly in the Results section, noting that the CoronaVac arm was excluded from this analysis due to insufficient immune response, as reported previously. |
||
|
Response 8: We thank the reviewer for this valuable comment. In the revised manuscript, we have clarified in the Results section that the CoronaVac booster arm was excluded from this analysis due to insufficient immune response and limited evaluable data, as previously reported in the earlier phase of the BCOV21 study.
Errors: 1. Lines 107, 135: Use the correct vaccine name ChAdOx1-S consistently throughout. 2. Line 196: The IgG anti-S-RBD outcome is a quantitative scale with an absolute zero, so negative values (e.g. −12 079.8, −4214.1) are impossible unless representing the lower bound of a 95 % CI. Please check these values again. 3. Temperature notation: Use the proper symbol: > 40 °C (rather than > 40oC). 4. Wording consistency: Replace “no statistical difference” with “no statistically significant difference” throughout. 5. Figure 1: Ensure consistent capitalisation and use ChAdOx1-S instead of Chadox1-S.
Response Error: We thank the reviewer for this careful observation, we have revised the vaccine name, the symbol of the temperature, no statistically difference, and the capitalization of ChAdOx1-S. As for the error no. 2: The negative IgG anti–S-RBD values in the results do not represent absolute concentrations but rather the difference between post-booster and pre-booster antibody levels. In some participants, baseline antibody concentrations were relatively high and showed a decline after vaccination, resulting in negative change values. |
||
|
|
||
Reviewer 2 Report
Comments and Suggestions for Authors
Strengths of this paper include robust randomization, clear immunogenicity endpoints, and integration with companion work (BCOV21). Although the paper offers a scientific interest, there are some significant comments:
- The sample size of 329 participants is sufficient for the primary immunogenicity endpoint; however, no power estimate is provided. Please include information on the expected effect size, power (e.g., 80–90%), and alpha level utilized to ascertain this sample size in the Statistical Analysis portion.
- The procedures for collecting data, such as blood samples on Days 0 and 28 and monitoring adverse events with diary cards, are well-explained. However, the choice of a smaller conventional NAb subsample (10 samples instead of 60 for sVNT) needs more explanation. Please explain the reason (for example, lack of resources) and talk about how this might affect generalizability.
- The application of geometric mean titers (GMTs) with 95% confidence intervals and non-parametric tests (Mann-Whitney, chi-square) is appropriate for immunogenicity data, which frequently display skewness; however, the fold-increase calculations (e.g., GMI for NAbs) could be enhanced to incorporate seroconversion rates stratified by baseline serostatus for more comprehensive analysis.
- The conclusion of no differences between half- and full-dose ChAdOx1-S groups (p=0.539 for GMT) is suitably cautious, but the discussion does not address the potential clinical significance of the difference in point estimates (3318 vs. 3743 AU/mL). It is advisable to incorporate effect size measures (e.g., Cohen's d) or non-inferiority margins to provide context for non-significance.
- Although adverse event reporting successfully classifies severity (mild predominant), it does not compute incidence rates per 100 person-days; these should be included in tables to support safety claims and make comparisons with previous research easier.
- The results section effectively illustrates baseline similarities (Table 1), but post-booster GMT changes are narratively explained without the use of forest plots or effect summaries; for visual clarity, it is suggested to include an additional figure that summarises GMT ratios (booster vs. baseline) across groups.
- While the conclusion that half-doses are "as immunogenic and safe as full doses" is overly general, it is based primarily on non-significant p-values; use confidence intervals to avoid implying equivalency in the absence of formal non-inferiority testing.
- Inclusion criteria specify healthy adults ≥18 years fully vaccinated with ChAdOx1-S 6–9 months prior, which aligns with the study's focus; however, detailed exclusion criteria (e.g., immunocompromised states, prior COVID-19 infection) are not explicitly listed—include a table or bulleted list of full criteria to allow replication.
- Although it makes sense, the exclusion of the CoronaVac booster arm from analysis because of its poor response in a previous study (BCOV21) should be clearly justified in the Methods section as a post-hoc decision. Any sensitivity analyses verifying that the exclusion did not skew the results should be reported.
- GMT comparisons provide strong evidence for the superior immunogenicity of BNT162b2 over ChAdOx1-S (p<0.001); however, interpretation should consider confounding by vaccine platform differences (mRNA vs. viral vector); subgroup analyses by age stratum are suggested to investigate heterogeneity.
Author Response
We would like to thank you for your valuable comments and constructive feedback on our manuscript. We have carefully revised the paper according to the reviewers’ and editors’ suggestions. All requested corrections, explanations, and supporting documents have been provided as detailed in our point-by-point response.
|
Response to Reviewer Comments
|
||
|
1. Summary |
|
|
|
Thank you very much for taking the time to review this manuscript. Please find the detailed responses below and in track changes in the re-submitted files.
|
||
|
2. Questions for General Evaluation |
Reviewer’s Evaluation |
Response and Revisions |
|
Does the introduction provide sufficient background and include all relevant references? |
Can be improved |
Thank you for your evaluation. |
|
Are all the cited references relevant to the research? |
Can be improved |
Thank you for your evaluation, we have updated the cited references. |
|
Is the research design appropriate? |
Can be improved |
Thank you for the review. We agree to the reviewers’ evaluation, and we also try to point out the limitations of this study. |
|
Are the methods adequately described? |
Can be improved |
Thank you for the critical review. We tried to give several details of the methodology section. |
|
Are the results clearly presented? |
Can be improved |
Thank you for this evaluation. We agree to the reviewers’ evaluation. |
|
Are the conclusions supported by the results? |
Can be improved |
Thank you for this evaluation. We agree to the reviewers’ evaluation. |
|
3. Point-by-point response to Comments and Suggestions for Authors |
||
|
Comments 1: The sample size of 329 participants is sufficient for the primary immunogenicity endpoint; however, no power estimate is provided. Please include information on the expected effect size, power (e.g., 80–90%), and alpha level utilized to ascertain this sample size in the Statistical Analysis portion.
|
||
|
Response 1: We thank the reviewer for the comment. We have revised the Statistical Analysis section.
|
||
|
Comments 2: The procedures for collecting data, such as blood samples on Days 0 and 28 and monitoring adverse events with diary cards, are well-explained. However, the choice of a smaller conventional NAb subsample (10 samples instead of 60 for sVNT) needs more explanation. Please explain the reason (for example, lack of resources) and talk about how this might affect generalizability. |
||
|
Response 2: We thank the reviewer for this valuable comment. It has been added to the Discussion section of the revised manuscript. |
||
|
|
||
|
Comments 3: The application of geometric mean titers (GMTs) with 95% confidence intervals and non-parametric tests (Mann-Whitney, chi-square) is appropriate for immunogenicity data, which frequently display skewness; however, the fold-increase calculations (e.g., GMI for NAbs) could be enhanced to incorporate seroconversion rates stratified by baseline serostatus for more comprehensive analysis. |
||
|
Response 3: We thank the reviewer for the constructive suggestion. The geometric mean increase (GMI) calculations have been described in detail in the Results section, specifically at lines 264, 274, 284, and 290 of the revised manuscript.
|
||
|
Comments 4: The conclusion of no differences between half- and full-dose ChAdOx1-S groups (p=0.539 for GMT) is suitably cautious, but the discussion does not address the potential clinical significance of the difference in point estimates (3318 vs. 3743 AU/mL). It is advisable to incorporate effect size measures (e.g., Cohen's d) or non-inferiority margins to provide context for non-significance |
||
|
Response 4: We thank the reviewer for this insightful comment. The effect size measures (Cohen’s d) and corresponding interpretation have been added to the Results section |
||
|
Comments 5: Although adverse event reporting successfully classifies severity (mild predominant), it does not compute incidence rates per 100 person-days; these should be included in tables to support safety claims and make comparisons with previous research easier
Response 5: We thank the reviewer for this valuable suggestion. The incidence rates of adverse events per 100 person-days have been calculated and added to the Results section
Comments 6: The results section effectively illustrates baseline similarities (Table 1), but post-booster GMT changes are narratively explained without the use of forest plots or effect summaries; for visual clarity, it is suggested to include an additional figure that summarises GMT ratios (booster vs. baseline) across groups. Response 6: We thank the reviewer for this helpful suggestion. We appreciate the value of a forest plot; however, we chose to present the post-booster GMT changes in detailed paragraphs and tables to clearly convey the numerical comparisons and maintain consistency with the overall presentation style of the manuscript. |
||
|
|
||
|
Comments 7: While the conclusion that half-doses are "as immunogenic and safe as full doses" is overly general, it is based primarily on non-significant p-values; use confidence intervals to avoid implying equivalency in the absence of formal non-inferiority testing. |
||
|
|
||
|
Response 7: We thank the reviewer for this valuable comment. The interpretation using confidence intervals and clarification regarding the absence of formal non-inferiority testing have been added to the Discussion section. |
||
|
|
||
|
Comments 8: Inclusion criteria specify healthy adults ≥18 years fully vaccinated with ChAdOx1-S 6–9 months prior, which aligns with the study's focus; however, detailed exclusion criteria (e.g., immunocompromised states, prior COVID-19 infection) are not explicitly listed—include a table or bulleted list of full criteria to allow replication. |
||
|
Response 8: We thank the reviewer for this helpful suggestion. The detailed inclusion and exclusion criteria have been added in Appendix A of the revised manuscript.
Comments 9: Although it makes sense, the exclusion of the CoronaVac booster arm from analysis because of its poor response in a previous study (BCOV21) should be clearly justified in the Methods section as a post-hoc decision. Any sensitivity analyses verifying that the exclusion did not skew the results should be reported.
Response 9: We thank the reviewer for this insightful comment. The justification for the post-hoc exclusion of the CoronaVac booster arm has been added to the Methods section of the revised manuscript.. |
||
|
Comments 10: GMT comparisons provide strong evidence for the superior immunogenicity of BNT162b2 over ChAdOx1-S (p<0.001); however, interpretation should consider confounding by vaccine platform differences (mRNA vs. viral vector); subgroup analyses by age stratum are suggested to investigate heterogeneity.
Response 10: We thank the reviewer for this valuable comment. The age distribution was similar across all study groups, and no elderly participants were included. Therefore, subgroup analyses by age stratum were not performed, as the sample size within each subgroup was limited and the study was not powered for stratified analysis |
||
Reviewer 3 Report
Comments and Suggestions for Authors
This manuscript investigates the effects of various COVID19 vaccines and doses on their effectiveness and adverse reactions. The data appear sound although the sample size appears to be relatively small. Importantly, the authors show that a half-dose booster is capable of generating a robust immune response although slightly weaker than a full dose vaccination. I have two comments to be addressed. First, i believe that authors are grossly wrong about the number of vaccines given since the start of the pandemic. They suggest the number is over 171 billion while my analysis of the literature suggests it is less than a tenth of that (on the order of 15 billion). Second, the authors indicate that this is part of a bigger study that was previously published. I believe the authors should better explain how this fits with the previous study and what exactly is new data.
Author Response
We would like to thank you for your valuable comments and constructive feedback on our manuscript. We have carefully revised the paper according to the reviewers’ and editors’ suggestions. All requested corrections, explanations, and supporting documents have been provided as detailed in our point-by-point response.
|
Response to Reviewer Comments
|
||
|
1. Summary |
|
|
|
Thank you very much for taking the time to review this manuscript. Please find the detailed responses below and in track changes in the re-submitted files.
|
||
|
2. Questions for General Evaluation |
Reviewer’s Evaluation |
Response and Revisions |
|
Does the introduction provide sufficient background and include all relevant references? |
Yes |
Thank you for your evaluation. |
|
Are all the cited references relevant to the research? |
Yes |
Thank you for your evaluation. |
|
Is the research design appropriate? |
Yes |
Thank you for the evaluation. |
|
Are the methods adequately described? |
Yes |
Thank you for the evaluation |
|
Are the results clearly presented? |
Yes |
Thank you for the evaluation. |
|
Are the conclusions supported by the results? |
Yes |
Thank you for the evaluation. |
|
3. Point-by-point response to Comments and Suggestions for Authors |
||
|
Comments: This manuscript investigates the effects of various COVID19 vaccines and doses on their effectiveness and adverse reactions. The data appear sound although the sample size appears to be relatively small. Importantly, the authors show that a half-dose booster is capable of generating a robust immune response although slightly weaker than a full dose vaccination. I have two comments to be addressed. First, i believe that authors are grossly wrong about the number of vaccines given since the start of the pandemic. They suggest the number is over 171 billion while my analysis of the literature suggests it is less than a tenth of that (on the order of 15 billion). Second, the authors indicate that this is part of a bigger study that was previously published. I believe the authors should better explain how this fits with the previous study and what exactly is new data.
|
||
|
Response: We thank the reviewer for these valuable comments. The global vaccination figure has been corrected based on updated WHO data (approximately 13.64 billion doses administered worldwide), and the Introduction section has been revised accordingly. In addition, we have added an explanation in the Methods section clarifying how this study fits within the larger BCOV21 trial and specifying which parts of the data are new and distinct from the previously published study.
|
||
Round 2
Reviewer 1 Report
Comments and Suggestions for Authors
Thank you for thoroughly addressing the concerns I raised in your previous submission.